# Can TP53, TMB and TME Expand the Immunotherapy Benefit in Metastatic Colorectal Cancer?

**DOI:** 10.3390/cancers17243984

**Published:** 2025-12-13

**Authors:** Monia Specchia, Denise Drittone, Eva Mazzotti, Federica Mazzuca

**Affiliations:** 1Medical Oncology Unit, Sant’Andrea Hospital in Rome, Via di Grottarossa 1035-1039, 00189 Rome, Italy; denise.drittone@uniroma1.it (D.D.); federica.mazzuca@uniroma1.it (F.M.); 2Oncology Unit, Department of Clinical and Molecular Medicine, Sapienza University of Rome, Via di Grottarossa 1035-1039, 00189 Rome, Italy

**Keywords:** metastatic colorectal cancer, TP53 mutation, immunotherapy, microsatellite instability, precision oncology, tumor biomarkers, immune checkpoint inhibitors

## Abstract

Colorectal cancer that has spread often does not respond to current immunotherapy. Many of these tumors harbor TP53 mutations, which are linked to aggressive behavior. We reviewed laboratory and clinical evidence to ask whether three features—TP53 status, the total number of DNA changes in the tumor (tumor mutational burden), and the surrounding tumor ecosystem (tumor microenvironment)—can guide more innovative use of immunotherapy. We outline how combining immune drugs with chemotherapy or medicines that normalize tumor blood vessels may help “open” resistant tumors, especially when the mutation burden is higher. Our goal is to provide a practical, precision approach that helps doctors select patients and treatment combinations earlier in care, beyond the small group already known to respond, and to highlight priorities for future trials. If validated, this strategy could improve outcomes for many people with metastatic colorectal cancer.

## 1. Introduction

Colorectal cancer (CRC) remains a major global health burden—third worldwide in incidence and second in cancer-related mortality—with approximately 1.9 million new cases and 900,000 deaths reported in 2020; these figures are projected to rise with population aging and the prevalence of modifiable risk factors [1]. Despite advances with cytotoxic doublet/triplet chemotherapy, anti-VEGF and anti-EGFR monoclonal antibodies, and refined surgical and interventional approaches, outcomes in metastatic CRC (mCRC) remain unsatisfactory; 5-year survival in stage IV disease is still low [2,3]. These observations underscore the need to identify molecular biomarkers that can guide management from the outset of care, moving beyond the simple dichotomy of microsatellite stability versus instability (MSS vs. MSI). The evidence summarized here supports a precision-medicine framework aimed at achieving deeper and more durable responses in molecularly selected patients, and at rigorously testing whether microsatellite-stable (MSS) tumors—traditionally considered poor candidates for immunotherapy—can nonetheless benefit, and through which underlying biological mechanisms. Consistent with this rationale, we prioritize mechanism-based therapeutic strategies and advocate for their earlier deployment in the first-line setting.

### 1.1. Rationale for Focusing on TP53

TP53 mutations occur in approximately half of colorectal cancers (CRCs) and are closely associated with chromosomal instability (CIN), resistance to genotoxic stress, and poorer prognosis [4,5]. In metastatic CRC (mCRC), TP53 alterations frequently co-occur with lesions in the RAS pathway, compounding therapeutic resistance and shaping evolutionary trajectories under treatment pressure [6]. Loss or mutation of p53 disrupts DNA-damage sensing, cell-cycle checkpoints, apoptosis, and tumor–microenvironment (TME) crosstalk, thereby influencing intrinsic tumor fitness and the likelihood of benefit from systemic therapies. TP53 should no longer be relegated to a purely prognostic aside; it represents a logical starting point for composite, precision-oriented decision-making.

### 1.2. The Current Immunotherapy Baseline

Immune checkpoint blockade (ICB) has transformed outcomes for the minority of patients with mismatch-repair–deficient/microsatellite-instability–high (dMMR/MSI-H) mCRC. In KEYNOTE-177, first-line pembrolizumab significantly prolonged progression-free survival versus chemotherapy and yielded durable overall survival [7]. Nivolumab, alone or with low-dose ipilimumab, demonstrated consistent activity in previously treated MSI-H/dMMR disease (CheckMate 142) [8]. However, these successes have not extended to the much larger population with mismatch-repair–proficient/microsatellite-stable (pMMR/MSS) tumors, where PD-1/PD-L1 monotherapy has limited efficacy; accordingly, guidelines endorse ICB as standard of care only for MSI-H/dMMR disease and recommend clinical-trial enrollment in most other settings [9,10].

### 1.3. Centrality of Tumor Mutational Burden (TMB)

Tumor mutational burden (TMB) integrates the cumulative imprint of genomic damage and correlates with neoantigen load and responsiveness to ICB across tumor types [11,12,13]. MSI-H tumors occupy the high end of the TMB spectrum, whereas most MSS CRCs—including many TP53-mutated cases—fall within low-to-intermediate ranges. Notably, a subset of MSS/TP53-mutated mCRC exhibits intermediate-to-high TMB that may suffice for immune recognition when combined with strategies that increase antigen visibility, reverse immune exclusion, and attenuate dominant immunosuppressive circuits. Framed in this way, TP53 is not merely a passenger but a driver of mutational processes that shape TMB, immunoediting, and ultimately sensitivity to ICB. Historically, clinical oncology has emphasized TP53’s prognostic and chemoresistance roles while paying comparatively less attention to its immunologic consequences—potentially obscuring opportunities precisely in patients whose TMB lies near an actionable threshold.

A major limitation in interpreting TMB as a biomarker in MSS CRC is the lack of cross-platform standardization. The absolute value of “high” TMB varies considerably among NGS panels due to differences in genomic footprint, bioinformatic pipelines, and filtering strategies. For example, FoundationOne, Guardant360 may classify the same tumor with differing TMB values [14,15]. Although ≥10 mut/Mb is frequently cited as a tumor-agnostic threshold, it has not been validated specifically in MSS CRC, nor is there a consensus cut-off for TP53-mutated MSS tumors. As such, the term “intermediate-to-high TMB” should be interpreted cautiously, and future harmonization efforts will be essential to establish panel-independent thresholds for clinical decision-making.

### 1.4. How TP53 Reshapes the TME

Loss of wild-type p53 can dampen antigen-presentation programs, favor PD-L1 upregulation, and recruit immunosuppressive myeloid and regulatory T-cell populations (MDSCs, TAMs, Tregs) within a hypoxic, VEGF- and TGF-β-enriched milieu [16,17,18]. The result is an immune-excluded or inflamed-yet-suppressed phenotype that is poorly permissive to PD-1/PD-L1 monotherapy. Conversely, TP53 loss may heighten replicative stress and cGAS–STING activation in certain contexts, potentially enhancing immunogenicity when paired with appropriate priming and vascular-normalizing partners—an effect most evident when panel-calibrated TMB is higher. TP53 mutation is not monolithic: conformational versus DNA-contact alterations, truncations, and compound events can exert distinct effects on cellular stress responses and immune signaling [4,5,6,18]. Clinically, these nuances matter because they modulate both the quantity of mutations (and thus putative neoantigens) and the quality of antigen presentation, helping explain why intermediate TMB may be sufficient for durable immune control in some MSS/TP53-mutated tumors but not others. Although TP53 mutations can be broadly categorized into conformational and DNA-contact classes, emerging evidence suggests that some recurrent hotspots (e.g., R175H, R248Q/W, R273H/C) generate shared neoantigens or exhibit gain-of-function programs that could theoretically influence tumor immunogenicity. However, to date, no CRC-specific studies have established a consistent association between individual TP53 hotspots and an immune-responsive phenotype. This represents a significant knowledge gap and an important area for future investigation.

### 1.5. Signals from Combination Studies

In unselected pMMR/MSS mCRC, adding atezolizumab to FOLFOXIRI plus bevacizumab (AtezoTRIBE) improved progression-free survival, suggesting that chemotherapy and anti-angiogenic “priming” can unlock ICB benefit in permissive biology [19]. By contrast, nivolumab plus mFOLFOX6/bevacizumab (CheckMate 9X8) and lenvatinib plus pembrolizumab (LEAP-017) did not meet their primary endpoints [20,21]. Complementary evidence from the randomized ROME trial shows that genomically matched, next-generation sequencing (NGS)-guided strategies can improve outcomes and, in select cases, nominate ICB for TMB-high MSS disease [22].

### 1.6. Where We Are Going and Why It Matters for Patients

The field is moving from binary eligibility (MSI-H vs. MSS) toward composite, biology-anchored selection that integrates TP53 status, calibrated TMB, and remediable TME features. For practicing oncologists, the implications are concrete: (i) prioritize early-line regimens that combine PD-1/PD-L1 agents with cytotoxic and anti-VEGF partners in biologically enriched settings; (ii) use dynamic pharmacodynamic readouts (e.g., early ctDNA kinetics, induction of interferon-stimulated gene signatures) to adapt therapy; and (iii) channel patients toward biomarker-anchored trials when single-agent ICB is unlikely to succeed [7,8,9,10,13,19,20,21,22]. Re-centering TMB and the immunologic consequences of TP53 in mCRC is not academic; it determines who can access effective immunotherapy now and where combination strategies can realistically extend benefit next.

### 1.7. Organotropism and ctDNA

Liver metastases in mCRC impose systemic immunologic tolerance through antigen-presenting cell sequestration and cytokine gradients that can blunt T-cell priming—a phenomenon increasingly recognized in prospective analyses of ICI-containing regimens. This organotropism intersects with TP53 and TMB: patients without dominant hepatic disease, or with well-controlled liver burden, appear more likely to benefit from chemo–anti-VEGF–PD-1/PD-L1 strategies when their mutational background is permissive. Growing evidence demonstrates that hepatic organotropism exerts a profound and quantifiable immunosuppressive effect in metastatic colorectal cancer. Across solid tumors—including CRC—patients with liver metastases experience a 40–60% reduction in objective response rates to PD-1/PD-L1 inhibitors compared with those without hepatic involvement, due to the liver’s capacity to induce systemic immune tolerance [23,24,25]. Mechanistically, hepatic metastases promote the sequestration of antigen-presenting cells, the expansion of tolerogenic macrophages, and impaired T-cell priming, thereby blunting systemic antitumor immunity. In parallel, several studies have characterized a myeloid-inflamed MSS CRC phenotype strongly associated with resistance to immune-checkpoint blockade. Mackenzie et al. [23] demonstrated that liver metastatic niches drive systemic T-cell deletion through myeloid-mediated mechanisms. Tauriello et al. [24] showed that TGF-β–rich stromal remodeling and myeloid exclusion create an immune-excluded TME that suppresses cytotoxic infiltration and reduces responsiveness to PD-1 blockade. Hegde and Chen [25] identified myeloid-dominant tumors as among the least responsive to ICI across cancers, due to multiple redundant suppressive pathways within the TME. Together, these quantitative and mechanistic insights indicate that MSS CRC with hepatic involvement and myeloid-inflamed TME represents a biologically distinct and highly ICI-resistant subgroup, in which strategies targeting stromal, vascular, or myeloid pathways may be required to overcome primary resistance.

Finally, dynamic biomarkers—especially early ctDNA decline and induction of interferon-stimulated gene programs—offer real-time confirmation that immune recognition is occurring, enabling escalation or de-escalation before radiographic changes [7,8,9,10,13,19,20,21,22]. These practical considerations underpin the precision-oncology approach advanced in this manuscript and justify a focused appraisal of specific evidence in mCRC.

## 2. Methods

### 2.1. Design and Quality Framework

This narrative, non-systematic review, was conducted in accordance with the SANRA criteria (Scale for the Assessment of Narrative Review Articles), with attention to justification of the topic, a focused aim, comprehensive literature coverage, transparent presentation, and appropriate interpretation [26]. A narrative, non-systematic design was deliberately chosen rather than a comprehensive systematic review because the current evidence base on TP53, TMB, and TME in metastatic colorectal cancer is highly heterogeneous, spanning early-phase clinical trials, biomarker-defined subgroups, and mechanistic or translational studies with inconsistent reporting of these biomarkers. Our primary objective was to provide a concept-driven, clinically oriented synthesis and to propose a precision-oncology framework, rather than to exhaustively catalogue all published reports or perform quantitative pooling. This narrative review was conducted with transparent reporting of: (i) the importance of the topic for the readership, (ii) clearly stated aims, (iii) the literature search, (iv) appropriate referencing, (v) balanced presentation of the available evidence, and (vi) appropriate presentation of relevant outcome data. In line with items 1 and 2, we first outlined the clinical relevance of TP53 alterations, tumor mutational burden (TMB), and the tumor microenvironment (TME) in metastatic colorectal cancer and formulated the aim of the review as a critical synthesis of how these biomarkers may influence immunotherapy benefit. Regarding item 3 (literature search), we searched PubMed and Scopus up to June 2024 using combinations of the keywords “TP53,” “colorectal cancer,” “microsatellite instability,” “tumor mutational burden,” “tumor microenvironment,” and “immunotherapy.” Reference lists of relevant reviews were also screened to identify additional studies. Articles were included if they discussed TP53 alterations in colorectal cancer and their relationship with immunotherapy outcomes, TMB, or TME features. Case reports, non-English-language papers, conference abstracts without full-text availability, and preclinical studies without clear translational relevance were excluded. Approximately 120 articles were screened and 78 were ultimately included in the narrative synthesis. In accordance with items 4–6, we aimed to provide consistent, up-to-date referencing, a critical, concept-driven synthesis of the evidence, and an appropriate qualitative presentation of outcome data from key clinical and translational studies, without performing formal meta-analysis or quantitative pooling.

### 2.2. Data Sources and Scope

We searched PubMed/MEDLINE, Scopus, and Web of Science for literature published from 2010 through 30 September 2025. To ensure clinical relevance and alignment with contemporary practice, we also reviewed the 2023 ESMO and 2025 NCCN guidelines and hand-searched reference lists of pertinent articles [9,10]. The review prioritizes studies in metastatic colorectal cancer (mCRC) but includes earlier-stage data when mechanistically informative (e.g., TP53 biology, TMB, TME features) and clearly applicable to immunotherapy strategies.

### 2.3. Search Strategy and Selection

The strategy used concept blocks combined with Boolean operators and database-specific subject headings/keywords for: colorectal cancer; TP53 or p53; microsatellite status (MSI/dMMR and MSS) and TMB; immunotherapy (including PD-1 and PD-L1 inhibitors); and tumor-microenvironment (TME) mechanisms relevant to TP53 (e.g., antigen processing/presentation, cytokine signaling, VEGF, TGF-β).

Two reviewers, MS and FM, independently screened titles/abstracts, followed by full-text assessment against predefined criteria. Eligible studies were clinical trials (phase I–III), real-world cohorts, systematic reviews/meta-analyses, and translational or preclinical investigations that directly examined TP53 status or function, MSI/TMB, TME correlates, or immune-checkpoint inhibition (ICI) in the context of colorectal cancer. We excluded non-English case reports, narrative commentaries without primary data, editorials, and studies with insufficient genomic or immunologic annotation to support the review aims. Preprints were considered only if a peer-reviewed version became available within the search window. Discrepancies were resolved by consensus.

### 2.4. Data Extraction and Synthesis

From each included report we abstracted study design, population and disease setting (with an emphasis on mCRC), TP53 assessment method and genotype (e.g., missense vs. truncating, hotspot status), MSI/dMMR vs. MSS classification, TMB measurement approach, TME readouts (e.g., antigen-presentation markers, PD-L1, myeloid/TGF-β signatures), intervention details (ICI alone or in combination), and key outcomes (objective response, PFS/OS, and—when available—ctDNA dynamics and translational endpoints).

### 2.5. Bias and Limitations

Risk of biased interpretation was mitigated through duplicate screening, structured data abstraction, and cross-checking of findings across study types. As a narrative review, this work was not prospectively registered and effect estimates should be interpreted cautiously where confounding and selection biases are likely.

## 3. Results

MSI-H as a benchmark for checkpoint blockade in mCRC. Microsatellite instability–high (MSI-H) disease provides a paradigm of immunotherapy success and anchors our framework for where—and why—checkpoint blockade can be effective in metastatic colorectal cancer (mCRC). In the phase III KEYNOTE-177 trial, pembrolizumab improved progression-free survival (PFS) versus chemotherapy (median 16.5 vs. 8.2 months; HR 0.60, 95% CI 0.45–0.79) and, with more than 5 years of follow-up, yielded a median overall survival (OS) of 77.5 vs. 36.7 months (HR 0.73, 95% CI 0.53–0.99), with 5-year OS rates of 54.8% vs. 44.2% despite extensive cross-over [7]. The durability and separation of survival curves reinforce MSI-H/dMMR mCRC as a distinctly immunotherapy-sensitive phenotype [27]. However, retrospective MSI-H/dMMR mCRC series treated with anti–PD-1/PD-L1 therapy have not systematically reported outcomes stratified by TP53 mutational status, and no consistent signal of an additive or synergistic effect of TP53 mutation within this intrinsically immunotherapy-sensitive subgroup has emerged to date; dedicated patient-level analyses are therefore needed [7,28].

Durability with dual checkpoint blockade. Durable activity has also been observed with nivolumab alone or with low-dose ipilimumab: in CheckMate 142, the ~4-year update reported an objective response rate of 65% (95% CI 55–73), complete responses of 13%, and median PFS and OS not reached [8]. Taken together, these data establish the biologic and clinical rationale for immunotherapy in MSI-H/dMMR mCRC and define a benchmark for comparison in microsatellite-stable disease [29].

TP53 as a contextual modifier in the broader CRC population. Against this benchmark, TP53 emerges as a key modifier of outcome in the broader CRC population. Mutations—present in approximately half of cases—show heterogeneous prognostic associations and, in specific contexts (including gain-of-function variants and co-mutation with KRAS), correlate with worse outcomes and reduced chemotherapy benefit [4,5,6]. In routine practice, patients with MSS/TP53-mutated tumors rarely receive immune checkpoint inhibitors outside clinical trials. Large genomic datasets place TP53 alterations predominantly within MSS disease, co-occurring with chromosomal instability and WNT/MAPK pathway events; these molecular constellations align with transcriptomic taxonomies that capture epithelial/proliferative and mesenchymal-like states with distinct immune ecologies [30,31].

TMB and the MSS/TP53 interplay. Tumor mutational burden (TMB) further modulates immunogenicity and response to PD-1/PD-L1 therapy across cancers [11,12,13]. While MSI-H usually entails high TMB, a fraction of MSS/TP53-mutated CRCs display intermediate-to-high TMB, suggesting potential for benefit when contextual biomarkers are favorable. Pan-tumor analyses consistently associate higher TMB with improved benefit from checkpoint blockade, while underscoring that tissue context and neoantigen clonality shape the predictive value of TMB [11,32]. In this direction, the randomized phase II ROME trial showed that genomically matched therapy outperformed standard care (PFS 3.5 vs. 2.8 months; HR 0.66, 95% CI 0.53–0.82; *p* = 0.0002) and improved objective responses (17.5% vs. 10%), supporting NGS-guided strategies that include immunotherapy for TMB-high tumors when appropriate [22].

Mechanistic basis for immune evasion in TP53-mutated CRC. Mechanistically, TP53 dysfunction reduces antigen visibility and promotes immune escape: wild-type p53 sustains antigen processing/presentation (e.g., TAP1/ERAP1) and represses PD-L1 via the p53–miR-34 axis, so TP53 loss can diminish MHC-I peptide loading and increase PD-L1 expression, weakening cytotoxic T-cell recognition [16,17,18]. These pressures are compounded by TP53-associated chromosomal instability, which generates micronuclei capable of activating cGAS–STING and innate sensing; in CRC, however, such signals are frequently buffered by a TGF-β–rich, myeloid-skewed, immune-excluded microenvironment that limits effector trafficking [23,24,33,34,35].

Therapeutic implications: converting “cold” to “hot”. Together, these results motivate combination strategies in MSS/TP53-mutated CRC aimed at converting “cold” tumors into “hot” ones—boosting antigenicity with DNA-damage-inducing chemotherapy or radiotherapy, restoring trafficking and function via vascular and stromal remodeling with anti-VEGF or TGF-β-pathway modulation, and addressing non-redundant inhibitory axes with dual-checkpoint approaches [25].

Conceptual synthesis. This framework explains why MSI-H tumors respond to PD-1 blockade and clarifies how, in selected MSS/TP53-mutated subgroups—particularly where TMB is intermediate-to-high and the microenvironment can be favorably reprogrammed—immunotherapy-containing regimens may deliver clinically meaningful benefit. Figure 1 provides a schematic summary of these concepts. A summary of key clinical and translational studies that underpin this TP53–TMB–TME framework is provided in Table 1.

## 4. Discussion

This narrative review evaluates the potential of immune checkpoint blockade (ICB) strategies in *TP53*-mutated colorectal cancer (CRC) and investigates the interplay among *TP53*, tumor mutational burden (TMB), and the tumor microenvironment (TME). Randomized studies consistently show that patients with microsatellite instability–high or mismatch-repair–deficient (MSI-H/dMMR) disease derive substantial and durable benefit from programmed cell death protein 1 (PD-1)-based therapy, whereas outcomes in mismatch-repair–proficient or microsatellite-stable (pMMR/MSS) disease remain modest and variable [7,8,19,20,21]. Stratifying MSI-H samples according to TMB and TME configuration is essential to elucidate potential interactions and their relevance to immunotherapy response. Among MSI-H tumors, TMB varies widely: high TMB values (>23 mut/Mb in colorectal cancer) are associated with an increased likelihood of response and prolonged survival with immune checkpoint inhibitors, whereas lower TMB correlates with early progression and reduced benefit [36,37,38,39,40,41]. However, not all MSI-H tumors display high TMB, and the combination of MSI-H status with high TMB identifies subgroups characterized by an immunoactive microenvironment and enriched neoantigen load [39,40] TME configuration—assessed through lymphocytic infiltration (CD8+, CD3+), PD-L1 expression, and inflammatory signatures—distinguishes MSI-H/high-TMB tumors (with high immune-cell density and IFNγ-driven responses) from MSI-H/low-TMB tumors (with a less inflamed TME and a higher fraction of genomic alterations) [39,40,41]. Co-occurring mutations, such as those in the Wnt pathway or biallelic MMR alterations, further modulate the TME and immune response [38,41]. In summary, stratification by TMB and TME in MSI-H tumors enables the identification of subgroups with differing immunogenicity and likelihood of response to immune checkpoint inhibitors, ultimately improving patient selection and therapeutic decision-making [36,37,39,40]. TP53 alterations are among the most common genomic events in CRC and frequently co-occur with canonical chromosomal instability (CIN) and MSS biology. Although TP53 loss can promote genomic instability, it does not consistently generate a high neoantigen burden. Most *TP53*-mutated CRCs are MSS with intermediate-to-low TMB and display an immune-excluded or myeloid-inflamed TME, resulting in limited responses to single-agent ICB and suboptimal outcomes with standard therapies.

Current combination strategies in pMMR/MSS CRC aim to transform a noninflamed TME into an antigen-presenting, inflamed milieu. For example, the phase II AtezoTRIBE trial demonstrated improved progression-free survival (PFS) with atezolizumab plus FOLFOXIRI and bevacizumab. By contrast, adding nivolumab to mFOLFOX6/bevacizumab in CheckMate 9X8 and lenvatinib plus pembrolizumab in LEAP-017 did not meet primary endpoints [19,20,21]. These findings suggest that benefit is confined to specific biological subsets, such as those with pre-existing interferon signaling, higher (yet MSS-range) TMB, immunogenic subclones, or a TME responsive to vascular endothelial growth factor (VEGF) or multikinase inhibition. It is important to acknowledge that, despite the biological rationale for ‘cold-to-hot’ conversion with chemotherapy and anti-VEGF or multikinase partners, large phase II–III trials such as CheckMate 9X8 and LEAP-017 did not meet their primary endpoints in unselected pMMR/MSS mCRC populations. At present, no published retrospective analyses from these studies have demonstrated a preferential benefit in TP53-mutated subgroups. Our hypothesis that TP53-mutated MSS tumors might be more amenable to such combinations is therefore based on mechanistic considerations (increased genomic stress, potential TMB enrichment, and a remodelable TME) rather than on definitive clinical evidence, and should be regarded as a testable framework for future biomarker-stratified trials rather than a proven effect. A precision-oncology strategy for *TP53*-mutated CRC should integrate genomic and microenvironmental features rather than rely on single biomarkers. Assessment of MSI/MMR status and panel-calibrated TMB can identify tumors with intrinsic antigenicity. Detection of *POLE/POLD1 exonuclease-domain mutations may indicate ultramutated, ICI-sensitive MSS disease. For the prevalent MSS/TP53-mutated group, evaluating transcriptomic and spatial features—such as interferon signaling, CD8 infiltration, and myeloid–stromal programs—can inform selection for appropriate ICB combinations, potentially with chemotherapy, anti-VEGF, or multikinase partners [7,8,19,20,21]. Therapeutic strategies should be adaptive, with pharmacodynamic markers—such as reductions in circulating tumor DNA (ctDNA) or induction of interferon-stimulated gene signatures—guiding escalation, maintenance, or de-escalation of ICB-containing regimens. This dynamic, biology-driven approach may improve disease control in previously poor-risk MSS/TP53-mutated subsets. Figure 2 presents a clinical algorithm for selecting optimal therapeutic strategies in *TP53*-mutated CRC.

Framed within precision medicine, these observations argue for composite, context-aware biomarkers that integrate TP53 alteration class (missense vs. truncating; hotspot; clonality), MSI/MMR status, panel-harmonized TMB, and spatial TME metrics to guide therapy from the earliest lines of care. Mechanistically, wild-type p53 sustains antigen processing/presentation and represses PD-L1 via the p53–miR-34 axis; conversely, TP53 loss can diminish MHC-I peptide loading and increase PD-L1, reducing cytotoxic T-cell recognition [31,32,33]. In parallel, TP53-associated CIN generates micronuclei that engage cGAS–STING and innate sensing, but these signals are frequently buffered by a TGF-β–rich, myeloid-skewed, immune-excluded microenvironment—clarifying why single-agent ICB underperforms in many MSS/TP53 tumors and why stromal/vascular modulation may be required [23,24].

In contemporary practice, this framework can be operationalized through baseline genotyping and longitudinal monitoring using tissue and liquid biopsy. Regulatory-cleared liquid assays such as Guardant360 CDx and FoundationOne Liquid CDx provide comprehensive genomic profiling (CGP) from plasma, enabling detection of targetable alterations, MSI status, and resistance mechanisms when tissue is unavailable or serial sampling is required [14,15]. Both platforms have undergone analytical and clinical validation, including U.S. regulatory premarket approval (PMA) for specific indications, supporting their use as adjuncts to tissue testing in advanced solid tumors. Beyond static profiling, ctDNA dynamics offer actionable pharmacodynamic readouts—early ctDNA clearance correlates with benefit and can inform escalation or de-escalation decisions, with growing evidence across CRC settings [42,43]. From a health-economics perspective, the value of liquid biopsy hinges on test price, turnaround time, and the extent to which biomarker-guided treatment reduces ineffective therapy and downstream costs; modeling suggests ctDNA-guided strategies can be cost-effective under realistic price thresholds and when they meaningfully alter chemotherapy utilization [44,45].

For MSS/TP53-mutated CRC specifically, we propose an iterative, biology-driven approach: (i) confirm MSS/MMR status and quantify TMB using harmonized panels; (ii) profile TP53 genotype class and clonality together with co-drivers (e.g., KRAS, BRAF, WNT/MAPK components); (iii) assess TME state via transcriptomic/spatial surrogates (interferon signaling, CD8 density, myeloid/TGF-β signatures); and (iv) integrate on-treatment ctDNA kinetics to refine the likelihood of durable benefit. Patients meeting composite thresholds—such as MSS with intermediate-to-high TMB, favorable interferon/T-cell infiltration, and modifiable stromal programs—may be prioritized for ICB-containing combinations with anti-VEGF or selected multikinase partners, whereas those lacking these features could be directed to alternative investigational strategies or cytotoxic backbones with biomarker-guided intensification. Prospective studies should therefore embed TP53-anchored composite biomarkers and prespecified translational endpoints, incorporate early radiographic plus ctDNA-based interim assessments, and stratify by MSI/TMB/TME context to de-risk negative trials and accelerate learning. Within this precision-oncology framework, the clinical algorithm in Figure 2 synthesizes how to triage TP53-mutated CRC into rational treatment paths, emphasizing readiness for early-line deployment when host and tumor immunity are most malleable. A practical limitation of the proposed ‘precise inclusion’ strategy is the integration of heterogeneous data streams—somatic DNA alterations (TP53, POLE, TMB), transcriptomic TME signatures, and spatial immune metrics—which are not yet routinely harmonized in standard clinical workflows. Implementing such composite biomarkers will likely require advanced clinical decision support systems (CDSS) capable of synthesizing multi-omic data for real-time interpretation. As medical informatics infrastructure evolves, these integrated pipelines will be essential to translate composite biomarker algorithms into everyday oncology practice.

## 5. Conclusions

This review reframes immunotherapy in colorectal cancer (CRC) from a binary paradigm to an integrated, biology-driven model in which TP53 functions as a contextual modifier alongside tumor mutational burden (TMB) and the tumor microenvironment (TME). In most CRCs, TP53 alteration aligns with CIN/MSS biology, lower antigenicity, and immune exclusion—features that help explain the modest activity of single-agent PD-1. Yet, within defined contexts—including co-occurring *POLE/POLD1 exonuclease-domain mutations, higher panel-calibrated TMB, and interferon-rich or angiogenesis-normalizable microenvironments—TP53 may contribute to identifying patients capable of mounting effective antitumor immunity. Building on these insights, our synthesis may support a shift from single-marker gatekeeping to composite, context-aware decision tools that integrate TP53 genotype class and clonality with MSI/MMR status, harmonized TMB thresholds, and spatially informed TME metrics.

The central clinical message is one of precision inclusion. Rather than excluding MSS/TP53-mutated disease from immunotherapy a priori, oncologists can triage patients based on integrated biology. When intrinsic antigenicity is present—because of ultramutation, clearly elevated TMB on assay-calibrated panels, or strong interferon signaling—PD-1 monotherapy remains a defensible choice. When trafficking barriers or stromal programs dominate—reflected by immune exclusion, myeloid skew, or angiogenic dependence—rational combinations (e.g., PD-1 with anti-VEGF or selected multikinase partners) become the preferred strategy. When antigen generation is limiting, cytotoxic or radiation priming can be sequenced to increase antigen release and presentation before or alongside checkpoint blockade. In each scenario, the role of TP53 is not to serve as a solitary predictor but to anchor the composite, connecting genotype with immune context and therapeutic design.

Equally important is a shift from static to dynamic decision-making. Baseline biology should be complemented by early on-treatment readouts—most notably circulating tumor DNA (ctDNA) clearance and induction of immune-activation signatures—that potentially provide real-time evidence of benefit. A practical algorithm can be envisioned: if early ctDNA declines substantially or immune-activation signatures rise, treatment may be maintained or de-escalated to minimize toxicity and preserve quality of life; if molecular response is absent despite adequate exposure, timely escalation or regimen modification could be pursued. This adaptive approach is particularly relevant in the first-line setting, when host and tumor immunity may be most amenable to reprogramming and the long-term cost of suboptimal choices is greatest.

Operationalization in routine care appears feasible. The initial evaluation should pair tissue-based testing with liquid biopsy when tissue is scarce, archival, or when longitudinal sampling is required. In practice, this means involves confirming MSI/MMR status; measuring TMB on a validated, panel-harmonized scale; and characterizing TP53 with sufficient granularity to distinguish missense from truncating events, identify hotspot residues, and comment on clonality where possible. Parallel assessment of TME state—via standardized transcriptomic surrogates and, where available, spatial profiling— may help define whether the dominant bottleneck is antigenicity, trafficking, or immune suppression. Embedding these data into tumor-board workflows enables consistent, reproducible decisions traceable to biology rather than to treatment habit.

The novelty of this framework may lie in rendering TP53 clinically actionable without reifying it as a single-gene switch. By placing TP53 at the center of a composite biomarker, we aim to bridge well-described mechanistic links—p53-dependent antigen presentation, PD-L1 regulation, genomic instability and innate sensing, and stromal/angiogenic programs—with concrete therapeutic choices. This connection clarifies why MSI-H tumors respond and provides a biologically coherent path to benefit for selected MSS/TP53-mutated patients who would otherwise be categorized as non-responders. The approach appears scalable: as new readouts become available (e.g., refined spatial metrics or improved measures of neoantigen clonality), they can be integrated into the composite without forcing wholesale changes in clinical workflow.

Standardization represents an immediate next step. For TP53, we advocate reporting that specifies variant class (missense vs. truncating) and hotspot status and, where feasible, incorporates clonality to estimate the fraction of tumor cells affected. For TMB, cross-panel calibration and CRC-appropriate thresholds are required to reduce misclassification at the margins; values should be interpreted alongside expected neoantigen quality and clonality rather than as a raw count alone. For the TME, we encourage a shift toward reproducible, clinically portable measures—interferon signaling, CD8 density, and myeloid/TGF-β programs—that can be prospectively applied in trials and, ultimately, embedded in decision support for routine practice. The same principles apply to ctDNA: assays should be standardized for minimal residual disease and for pharmacodynamic response, with clear definitions of meaningful decline and guidance on how those changes should alter management.

Prospective validation remains the logical proving ground. Trials should stratify by *TP53*-anchored composite criteria; include early radiographic and molecular checkpoints; and predefine rules for escalation, maintenance, or de-escalation linked to on-treatment biology. Early-line studies are expected to yield the greatest benefit, both because immune competence is higher and because successful reprogramming can compound across subsequent lines. Complementary real-world evidence—through registry-based trials and structured learning-health-system datasets—could accelerate external validation and identify subgroups that may be under-represented in traditional studies. Together, these efforts will clarify which combinations deliver durable control with acceptable toxicity and in which biologic niches they should be deployed.

Implementation will depend on practicality and equity. Turnaround time, cost, and access determine whether composite biomarker strategies can be delivered outside major centers. Where resources are constrained, a tiered approach may prioritize tests with the greatest immediate impact on decision-making (e.g., MSI/MMR and TMB on a validated panel), adding layers—such as transcriptomic TME surrogates or spatial analysis—when they are likely to change management. Clear reporting templates and shared decision-making with patients can further align therapeutic choices with values and expectations, especially when trade-offs between efficacy, toxicity, and logistics are clinically relevant.

From a public health and health economics perspective, precision stratification using early liquid biopsy testing may also reduce downstream costs by preventing exposure to later-line treatments such as TAS-102 or regorafenib, which are expensive and provide limited survival benefit. Identifying likely responders earlier not only improves clinical outcomes but also represents a cost-effective allocation of resources, strengthening the rationale for tiered precision-testing approaches in resource-limited settings.

Taken together, the evidence suggests a transition from exclusion to informed inclusion for MSS/TP53-mutated CRC. By aligning mechanisms with therapy—and by measuring early whether the chosen mechanism appears to be working—clinicians can convert a historically poor-risk category into stratified, treatable subgroups. The goal is not merely incremental benefit but deeper, more durable responses for a larger proportion of patients, achieved with greater predictability and fewer cycles of ineffective treatment. If adopted, this precision framework could reshape how immunotherapy is used in metastatic CRC and, more broadly, how complex molecular signals are translated into routine, patient-centered care.

## Figures and Tables

**Figure 1 cancers-17-03984-f001:**
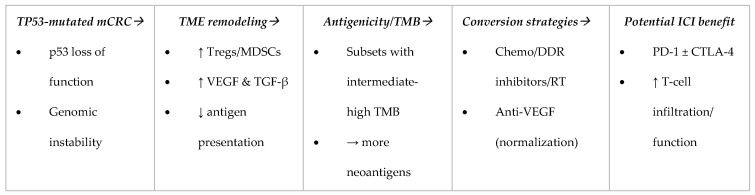
TP53-mutated mCRC: left-to-right flow showing how p53 loss reshapes the tumor microenvironment (TME) and, in subsets with higher tumor mutational burden (TMB), enables benefit from immune checkpoint inhibitors (ICIs) when paired with conversion strategies (chemotherapy/DDR inhibitors/radiotherapy and anti-VEGF). ↑: Increase ↓: Decrease.

**Figure 2 cancers-17-03984-f002:**
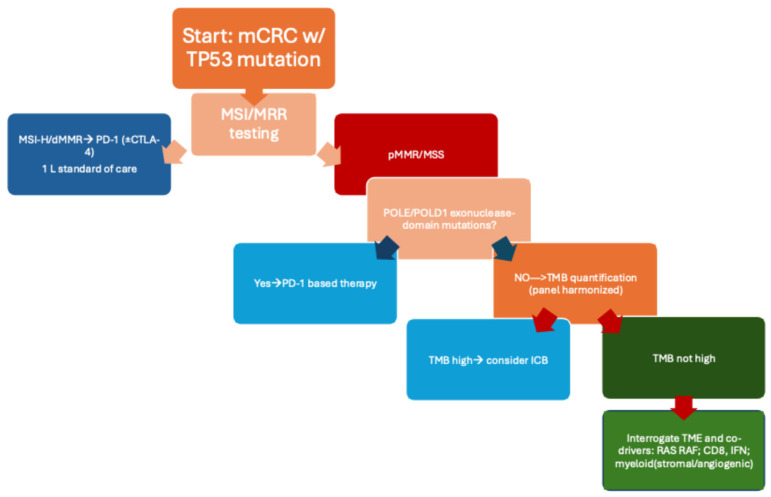
Precision selection flowchart for TP53-mutated CRC. Ordered decision nodes: MSI/MMR → POLE/POLD1 → TMB → co-drivers/TME to nominate PD-1 therapy or rational combinations in MSS disease.

**Table 1 cancers-17-03984-t001:** Evidence Summary Matrix of Key Immunotherapy Trials in Metastatic Colorectal Cancer (mCRC), Including MSI/MMR Status, TP53 Information, TMB Levels, TME Features, and Clinical Outcomes.

*Trial/Study*	*Population*	*MSI/MMR Status*	*TP53 Info*	*TMB Range*	*TME Features*	*ICI/Treatment Arms*	*Key Outcomes*	*Key References*
KEYNOTE-177	First-line mCRC	MSI-H/dMMR	Not stratified	High TMB (>23 mut/Mb typical)	Inflamed phenotype, high CD8	Pembrolizumab vs. chemotherapy	Improved PFS & OS; durable responses	[7,27]
CheckMate-142	Pre-treated mCRC	MSI-H/dMMR	Not stratified	High TMB	Inflamed TME	Nivolumab ± Ipilimumab	High ORR (65%), long-term PFS/OS	[8,29]
AtezoTRIBE	First-line mCRC	MSS/pMMR (majority)	Not reported	Low–intermediate TMB	Partial TME conversion with priming	FOLFOXIRI+Bev ± Atezolizumab	PFS improvement in unselected MSS subset	[19]
CheckMate-9X8	First-line mCRC	MSS/pMMR	Not reported	Low–intermediate TMB	Non-inflamed MSS TME	mFOLFOX6+Bev ± Nivolumab	No significant PFS benefit	[20]
LEAP-017	Pre-treated mCRC	MSS/pMMR	Not stratified	Low–intermediate TMB	Myeloid-inflamed, ICI-resistant	Pembrolizumab + Lenvatinib	No OS benefit vs. SOC	[21]
ROME Trial	Advanced solid tumors (CRC subset)	Mixed MSI & MSS	TP53 via NGS	TMB-high vs. TMB-low	Genomic clusters (not full TME)	Genomically matched therapy incl. ICI	Improved PFS; benefit in TMB-high MSS	[22]

## Data Availability

Data are cited in the text.

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
