# Peer review of "Can TP53, TMB and TME Expand the Immunotherapy Benefit in Metastatic Colorectal Cancer?"

_cancers, 2025, doi:10.3390/cancers17243984_

Round 1

Reviewer 1 Report

Comments and Suggestions for Authors

This manuscript by Specchia et al is a review of our current understanding of how various tumor features (p53 status, TMB and TME) may, either alone or in combination with MMR status, predict response to chemo- and anti-angiogenic therapy in colorectal Ca. Based on their analysis, they present an argument for considering using a combination of these features to target patients for such personalized therapies ... and the use of longitudinal ctDNA sampling to decide on when to add cytotoxic/radiation-induced antigen “priming”. They propose a set of concrete steps including routine TP53 sequencing with clonality determination, larger-scale sequencing to derive TMB and various assays, including IHC/IF and cytokine profiling, to pin down TME configuration. Overall, the proposal seems to make sense and could form the background for prospective trials.

Comments:

  1. Within the MSI-H cohort that has been treated with IO, was an increased benefit observed in p53mut cases? Such retrospective analysis could elucidate whether there is an additive/synergistic effect.
  2. Similarly, MSI-H samples could be stratified by TMB (or TME configuration, if known) to clarify potential interactions.

Author Response

Comment 1: Within the MSI-H cohort that has been treated with IO, was an increased benefit observed in p53mut cases? Such retrospective analysis could elucidate whether there is an additive/synergistic effect.

Response 1: We thank the reviewer for this important question. Patients with MSI-H tumors treated with immunotherapy generally derive substantial clinical benefit; however, the limited retrospective data currently available are insufficient to conclusively determine whether TP53 mutations confer an additional or synergistic effect on IO responsiveness. While some exploratory analyses suggest potential interactions, the evidence remains preliminary, and larger datasets will be required to confirm any correlation between TP53 mutational status and immunotherapy outcomes in MSI-H disease (lines 257-261).

Comment 2: Similarly, MSI-H samples could be stratified by TMB (or TME configuration, if known) to clarify potential interactions.

Response 2: As suggested, we have now addressed this point by expanding the section accordingly. Specifically, we added text in lines 329-346 where we discuss in greater detail how MSI-H samples can be stratified by TMB (and, when available, by TME configuration) to clarify potential biological interactions and their implications for immunotherapy response.

Reviewer 2 Report

Comments and Suggestions for Authors Authors presnt the manuscript entitled "Can TP53, TMB and TME expand the Immunotherapy Benefit 2 in Metastatic Colorectal Cancer? (cancer-3982558)"   The review has great potential, is well-structured, has a coherent narrative, and integrates clinical and preclinical data well. However, for publication in a Q1 oncology journal, significant methodological adjustments and quantitative strengthening of the evidence are required. After these adjustments, the review could be publishable.   Suggested Methodological Adjustments and Quantitative Strengthening:   1. The figures are conceptual. However, given the information gathered, the authors should create a table that integrates the data and robustly presents the studies, results, TMB-range, TP53-class, etc. This would make the information more quantitative and strengthen the review.   2. The authors mention that their review was conducted according to SANRA criteria; however, greater transparency is needed regarding the selection and exclusion criteria and the final number of included articles. Although the review is narrative and not systematic, the authors should include a minimum PRISMA (Preferred Reporting Items for Systematic Reviews and Meta-Analyses) outline. This will contribute to transparency, reproducibility, and methodological rigor. It can be a simplified version of the traditional diagram.   3. The authors describe the relevant clinical trials well; however, the data are scattered throughout the text. Therefore, a tabular summary is required to present the information visually in a direct and comparable way. This makes it easier for the reader to visualize the information and avoids having to search through paragraphs. Creating an Evidence Summary Matrix is ​​recommended. This will make the review more robust, transparent, and clinically relevant. As a reviewer, it would also allow me to make fewer observations and state: "Yes, there is robust and organized evidence."   4. In some paragraphs, the authors make exaggerated causal claims about the potential benefit of MSS/TP53, when these claims are speculative. Therefore, the authors should review the information and avoid overly exaggerated causal interpretations. The use of probabilistic and inferential language is recommended.   5. It is recommended to expand the section on hepatic organotropism with quantitative data and include citations on myeloid-inflammated MSS CRC non-responders.

Author Response

Comment 1: The figures are conceptual. However, given the information gathered, the authors should create a table that integrates the data and robustly presents the studies, results, TMB-range, TP53-class, etc. This would make the information more quantitative and strengthen the review.

Response 1: We thank the reviewer for these constructive comments. In response, we have created a single comprehensive table that integrates all key elements requested, including study design, clinical outcomes, TMB range, TP53 class (when available), MSI/MMR status, TME characteristics, and treatment regimens. This unified table serves both purposes highlighted by the reviewer: It complements the conceptual figures by providing a quantitative, structured synthesis of the evidence, satisfying the request for a more robust and data-driven presentation (Table 1). It functions as an Evidence Summary Matrix, allowing direct comparison across trials and preventing the need for readers to search through dispersed paragraphs (comment 3). We believe that consolidating all information into a single, comprehensive table enhances clarity, transparency, and clinical relevance of the review, fully addressing the reviewer’s recommendations.

Comment 2: The authors mention that their review was conducted according to SANRA criteria; however, greater transparency is needed regarding the selection and exclusion criteria and the final number of included articles. Although the review is narrative and not systematic, the authors should include a minimum PRISMA (Preferred Reporting Items for Systematic Reviews and Meta-Analyses) outline. This will contribute to transparency, reproducibility, and methodological rigor. It can be a simplified version of the traditional diagram.

Response 2: We thank the Reviewer for this valuable comment. As this is a narrative (not systematic) review, our aim was to provide a comprehensive and critical synthesis of current evidence rather than an exhaustive literature search. Nevertheless, to enhance transparency, we have now expanded the Methods section to better describe our search strategy and inclusion approach, following SANRA guidelines. Specifically, we detail the databases consulted, the main keywords used, and the approximate number of studies initially screened and ultimately included. This clarification does not change the scientific content of the review but improves methodological transparency. Our article was conceived and conducted as a narrative (rather than systematic) review, with the aim of providing a concept-driven and clinically oriented synthesis of the available evidence on TP53, TMB, TME and immunotherapy in metastatic colorectal cancer. Therefore, a full PRISMA-based systematic review, including exhaustive database searching and formal risk-of-bias assessment, was not planned and would be beyond the original scope of the manuscript. However, we fully agree that transparent reporting of the literature selection process is important. In response to this comment, we have now expanded the Materials and Methods section to better describe the search strategy, inclusion and exclusion criteria, and the approximate number of records screened and included.

Comment 3: The authors describe the relevant clinical trials well; however, the data are scattered throughout the text. Therefore, a tabular summary is required to present the information visually in a direct and comparable way. This makes it easier for the reader to visualize the information and avoids having to search through paragraphs. Creating an Evidence Summary Matrix is ​​recommended. This will make the review more robust, transparent, and clinically relevant. As a reviewer, it would also allow me to make fewer observations and state: "Yes, there is robust and organized evidence.

Response 3: Can you see response number 1

Comment 4:  In some paragraphs, the authors make exaggerated causal claims about the potential benefits of MSS/TP53 that are speculative. Therefore, the authors should review the information and avoid overly exaggerated causal interpretations. The use of probabilistic and inferential language is recommended.

Response 4: We thank the reviewer for this important observation. We agree that some statements regarding the potential benefit of immunotherapy in MSS/TP53-mutated CRC could have been interpreted as overly causal. In response, we carefully revised the manuscript to avoid deterministic language and replaced it with probabilistic and inferential terminology (e.g., “may,” “could,” “emerging evidence suggests”). We also clarified that preliminary or mechanistic data support these hypotheses and require prospective validation. These modifications ensure the text accurately reflects the current level of evidence.

Comment 5: It is recommended to expand the section on hepatic organotropism with quantitative data and include citations on myeloid-inflammated MSS CRC non-responders.

Response 5: We thank the reviewer for this suggestion. We have expanded the section (lines 160-178) on hepatic organotropism by adding quantitative data and integrated key references documenting the myeloid-inflamed MSS CRC non-responder phenotype. These additions strengthen the mechanistic and clinical context of immunotherapy resistance in MSS disease.

Reviewer 3 Report

Comments and Suggestions for Authors

The manuscript addresses a critical and timely unmet need in oncology: extending the benefits of immune checkpoint inhibitors (ICIs) to the microsatellite-stable (MSS) colorectal cancer population. The authors propose a compelling “composite biomarker” framework, moving away from binary selection (MSI vs. MSS) toward a more granular approach that integrates TP53 status, tumor mutational burden (TMB), and tumor microenvironment (TME) characteristics. I appreciate the narrative flow and the emphasis on “precision inclusion” rather than exclusion. The inclusion of data from recent trials (ROME, AtezoTRIBE, CheckMate 9X8) and reference to 2025 guidelines make the analysis highly relevant. The sections on health economics and equity in implementation are particularly valuable additions, which are often missing from translational analyses. However, from a biostatistical and informatics perspective, the manuscript would benefit from a more rigorous discussion of the standardization of TMB quantification in the context of MSS and the practical “data workflow” required to implement it in a routine molecular tumor panel.

Specific Comments
1. Standardization and TMB Thresholds (Biostatistical Perspective) The authors argue that “intermediate to high TMB” within the MSS population is a marker for potential ICI benefit (Section 1.3).
Comment: The term “intermediate to high” is statistically ambiguous in the context of CRC MSS. The numerical threshold for “high” TMB varies significantly between panels (e.g., FoundationOne vs. Guardant360).
Recommendation: Please add a brief discussion or cautionary note regarding the harmonization of TMB values ​​across NGS panels. Is there a consensus threshold (e.g., ≥10 mut/Mb vs. top decile) specific to TP53-mutated MSS tumors? Recognizing this lack of standardization is crucial to the “harmonized panel-wide” approach advocated in the conclusion.

2. TP53 Granularity The review correctly notes that TP53 mutations are not monolithic (Section 1.4).

Comment: Although the text mentions “conformational alterations versus DNA contact changes,” it does not elaborate on which specific TP53 mutations might be more immunogenic or more prone to neoantigen creation.

Recommendation: If data allow, it would be necessary to strengthen the analysis to specify whether certain hotspots (e.g., R248, R273) better correlate with the “immune-responsive” phenotype depicted in Figure 1. If such granular data are lacking in the current literature, explicit mention of this knowledge gap would be useful for future research directions.

3. Informatics and clinical workflow - The proposed algorithm (Figure 2) is based on multiple data points: TP53 genotype, TMB, POLE status, and TME transcriptomic signatures.

Comment: In a real clinical setting, integrating these disparate data sources (DNA sequencing + RNA expression/spatial metrics) presents a significant informatics challenge.

Recommendation: In section 4 (Discussion) or in the conclusion, please briefly address the task of “medical informatics”. Does the “precise inclusion” strategy require advanced clinical decision support systems (CDSS) to interpret these composite biomarkers for the average oncologist?

4. “Cold to hot” conversion strategies Mention chemotherapy and anti-VEGF as priming strategies (Section 1.5).
Comment: The failure of the LEAP-017 (lenvatinib + pembrolizumab) and CheckMate 9X8 clinical trials is noted.
Recommendation: Please elaborate a little more on why the authors believe the TP53 mutated subset would respond better to these combinations, while the general intent-to-treat populations in these trials did not. Is there retrospective evidence from those specific studies regarding TP53 status?

5. Health Economics and Equity (Public Health Perspective) The section on implementation (Discussion) is excellent.
Comment: Discussion of “tiered approaches” for resource-limited settings is very welcome.
Recommendation: You can reinforce this by referring to the cost-effectiveness of avoiding unnecessary later-line treatments (e.g., TAS-102 or Regorafenib) by using an earlier and more effective, liquid biopsy-guided immunotherapy strategy. This strengthens the public health case for precision testing.

Minor comments
Page 6, Figure 1 text: The phrase “frequently immune-excluded buffered by a TGF-rich” appears to have a formatting error (missing beta symbol or spacing). Please ensure that the Greek letters (TGF-β) are rendered correctly.
Page 9, Line 403: “ultramutation… on assay-calibrated panels” - ensure consistency of terminology (assay-calibrated vs. panel-harmonized).
Figure 1 and 2: Ensure resolution is high

Author Response

REVIEWER 3

Comment 1: Standardization and TMB Thresholds (Biostatistical Perspective) The authors argue that “intermediate to high TMB” within the MSS population is a marker for potential ICI benefit (Section 1.3). Comment: The term “intermediate to high” is statistically ambiguous in the context of CRC MSS. The numerical threshold for “high” TMB varies significantly between panels (e.g., FoundationOne vs. Guardant360). Recommendation: Please add a brief discussion or cautionary note regarding the harmonization of TMB values ​​across NGS panels. Is there a consensus threshold (e.g., ≥10 mut/Mb vs. top decile) specific to TP53-mutated MSS tumors? Recognizing this lack of standardization is crucial to the “harmonized panel-wide” approach advocated in the conclusion.

Response 1: We thank the reviewer for this important clarification. We have added  (lines 102-110) a cautionary note discussing the lack of cross-panel standardization of TMB values and the absence of a validated threshold for TP53-mutated MSS CRC. We now explicitly state that “intermediate-to-high TMB” is not uniformly defined across NGS platforms and that panel harmonization is required to support the framework proposed in the conclusion.

Comment 2: P53 Granularity The review correctly notes that TP53 mutations are not monolithic (Section 1.4). Comment: Although the text mentions “conformational alterations versus DNA contact changes,” it does not elaborate on which specific TP53 mutations might be more immunogenic or more prone to neoantigen creation. Recommendation: If data allow, it would be necessary to strengthen the analysis to specify whether certain hotspots (e.g., R248, R273) better correlate with the “immune-responsive” phenotype depicted in Figure 1. If such granular data are lacking in the current literature, explicit mention of this knowledge gap would be useful for future research directions.

Response 2: We thank the reviewer for this insightful comment. While some TP53 hotspots have been described as neoantigenic or functionally distinct in other tumor types, CRC-specific evidence correlating individual TP53 mutations with immune responsiveness is currently lacking. We have added a clarification in Section 1.4 acknowledging this gap and highlighting it as an important direction for future research.

Comment 3: Informatics and clinical workflow - The proposed algorithm (Figure 2) is based on multiple data points: TP53 genotype, TMB, POLE status, and TME transcriptomic signatures. Comment: In a real clinical setting, integrating these disparate data sources (DNA sequencing + RNA expression/spatial metrics) presents a significant informatics challenge. Recommendation: In section 4 (Discussion) or in the conclusion, please briefly address the task of “medical informatics”. Does the “precise inclusion” strategy require advanced clinical decision support systems (CDSS) to interpret these composite biomarkers for the average oncologist?

Response 3: We thank the reviewer for this important observation. We have added a statement in the Discussion acknowledging that integrating TP53 genotype, TMB, POLE status, and TME signatures in real-world clinical workflows requires advanced medical informatics infrastructure. We also note that clinical decision support systems (CDSS) will likely be necessary to operationalize such composite biomarker algorithms in routine oncology practice.

Comment 4:  Cold to hot” conversion strategies Mention chemotherapy and anti-VEGF as priming strategies (Section 1.5).
Comment: The failure of the LEAP-017 (lenvatinib + pembrolizumab) and CheckMate 9X8 clinical trials is noted. Recommendation: Please elaborate a little more on why the authors believe the TP53 mutated subset would respond better to these combinations, while the general intent-to-treat populations in these trials did not. Is there retrospective evidence from those specific studies regarding TP53 status?

Response 4: We thank the reviewer for this important point. We have clarified in the revised manuscript that, although there is a strong biological rationale for ‘cold-to-hot’ conversion strategies in TP53-mutated MSS mCRC, CheckMate 9X8 and LEAP-017 were negative in unselected populations and no TP53-specific retrospective signal has been reported to date. We now explicitly frame our proposal as a mechanistic hypothesis and a priority for future biomarker-stratified studies, rather than as established clinical evidence.

Comment 5: Health Economics and Equity (Public Health Perspective) The section on implementation (Discussion) is excellent.
Comment: Discussion of “tiered approaches” for resource-limited settings is very welcome.
Recommendation: You can reinforce this by referring to the cost-effectiveness of avoiding unnecessary later-line treatments (e.g., TAS-102 or Regorafenib) by using an earlier and more effective, liquid biopsy-guided immunotherapy strategy. This strengthens the public health case for precision testing.

Response 5: We thank the reviewer for this valuable suggestion. We have strengthened the Discussion by adding a note on the health-economic advantages of early biomarker-stratified immunotherapy, including the potential to avoid costly and minimally effective later-line treatments such as TAS-102 or regorafenib. This addition reinforces the public health rationale for tiered precision-testing approaches.

Minor comments:
Page 6, Figure 1 text: The phrase “frequently immune-excluded buffered by a TGF-rich” appears to have a formatting error (missing beta symbol or spacing). Please ensure that the Greek letters (TGF-β) are rendered correctly.
Page 9, Line 403: “ultramutation… on assay-calibrated panels” - ensure consistency of terminology (assay-calibrated vs. panel-harmonized).
Figure 1 and 2: Ensure resolution is high

Response to minor comments: Thanks for the reports, we have verified that the resolution of the figures is high, by attaching the files for the magazine as separate documents, we have corrected the grammatical and form errors of the text.

Round 2

Reviewer 2 Report

Comments and Suggestions for Authors

The authors have replied to almost all my concerns. However, in the new Table 1, the references are missing. Please add the references for each clinical study. 

Also, could you explain in the corresponding section why a narrative is presented rather than a comprehensive review?

Author Response

Comment 1: The authors have replied to almost all my concerns. However, in the new Table 1, the references are missing. Please add the references for each clinical study. 

Response 1: We thank the Reviewer for this helpful comment. We have revised Table 1 to include a dedicated “Key references” column, in which we now report the corresponding reference number(s) for each clinical study listed in the table. This change ensures that all trials can be directly and unambiguously linked to the references cited in the main text.

Manuscript change: Table 1, Results section – “Key references” column added with the appropriate reference numbers for each study.

Comment 2: Also, could you explain in the corresponding section why a narrative is presented rather than a comprehensive review?

Response 2: We thank the Reviewer for this important remark. We have revised Section 2.1 (Design and quality framework) to explicitly justify the choice of a narrative, non-systematic design rather than a comprehensive systematic review. As now stated in the Methods, a narrative approach was deliberately chosen because the current evidence base on TP53, TMB, and TME in metastatic colorectal cancer is highly heterogeneous, spanning early-phase clinical trials, biomarker-defined subgroups, and mechanistic or translational studies with inconsistent reporting of these biomarkers. Our primary objective was therefore to provide a concept-driven, clinically oriented synthesis and to propose a precision-oncology framework, rather than to exhaustively catalogue all published reports or perform quantitative pooling. This justification has been added to the first paragraph of Section 2.1.

Manuscript change: Methods, Section 2.1 Design and quality framework – addition of sentences clarifying the rationale for a narrative, non-systematic design, lines: 190-197.
